# Atomically precise gold nanoclusters at the molecular-to-metallic transition with intrinsic chirality from surface layers

Li-Juan Liu[1], Fahri Alkan[2], Shengli Zhuang[1,3], Dongyi Liu[1], Tehseen Nawaz[1], Jun Guo[1], Xiaozhou Luo [4] & Jian He [1,3] ✉

The advances in determining the total structure of atomically precise metal nanoclusters have prompted extensive exploration into the origins of chirality in nanoscale systems. While chirality is generally transferrable from the surface layer to the metal−ligand interface and kernel, we present here an alternative type of gold nanoclusters (138 gold core atoms with 48 2,4-dimethylbenzenethiolate surface ligands) whose inner structures are not asymmetrically induced by chiral patterns of the outermost aromatic substituents. This phenomenon can be explained by the highly dynamic behaviors of aromatic rings in the thiolates assembled via π − π stacking and C − H⋯π interactions. In addition to being a thiolate-protected nanocluster with uncoordinated surface gold atoms, the reported $Au_{138}$ motif expands the size range of gold nanoclusters having both molecular and metallic properties. Our current work introduces an important class of nanoclusters with intrinsic chirality from surface layers rather than inner structures and will aid in elucidating the transition of gold nanoclusters from their molecular to metallic states.

Chirality is prevalent in nature, from subatomic particles to enormous living systems. The development of chiral compounds and materials with important functions has a tremendous impact on daily life. In recent years, research on chiral metal nanoparticles has become increasingly significant due to their potent rotatory optical activity[1,2], excellent biocompatibility[3–5], and pronounced asymmetric induction capability[6–8] for a variety of optical, biomedical, and catalytic applications. Similar to chiral organometallic complexes, atomically precise metal nanoclusters, which are ultrasmall nanoparticles with a diameter of 1–3 nm, could be coordinated by chiral ligands to exhibit intriguing optical properties[9,10]. Soon after the structures of gold nanoclusters were revealed by single-crystal X-ray diffraction (SCXRD)[11,12], it was realized that the nanoclusters' chirality might also be associated with the staple distributions or kernel structures[13,14], rather than solely being a result of chirality transfer from the external chiral organic

components (e.g., chiral thiols[15], alkynes[16], phosphines[17], and N-heterocyclic carbenes[18]).

Extensive studies on chiral metal nanoclusters bearing achiral ligands have allowed for a profound grasp of their intrinsic chirality[19,20]. In the cases of thiolate-protected gold nanoclusters, the chiral arrangements of carbon terminal groups in the surface layer could have a strong "outside-in" influence on the distributions of −S−Au−S− motifs in the interfacial layer, giving rise to the transfer of chirality from the chiral surface to the interfacial layer or even the transitional layer of the kernel[13]. Therefore, all previously reported gold nanoclusters with chiral whirls of carbon tails, such as $Au_{133}(SR)_{52}$[21,22], $Au_{144}(SR)_{60}$[23], and $Au_{246}(SR)_{80}$[24], offer chiral patterns of −S−Au−S− motifs in the thiolate ligand assembly. However, since the carbon tails are mainly assembled via weak intra-cluster interactions[24], it is possible to observe interconversion between

[1]Department of Chemistry, The University of Hong Kong, Hong Kong, China. [2]Department of Nanotechnology Engineering, Abdullah Gül University, Kayseri, Turkey. [3]State Key Laboratory of Synthetic Chemistry, The University of Hong Kong, Hong Kong, China. [4]Center for Synthetic Biochemistry, Shenzhen Institute of Synthetic Biology, Shenzhen Institutes of Advanced Technology, Chinese Academy of Sciences, Shenzhen, China. ✉e-mail: jianhe@hku.hk

two enantiomeric forms of the chiral surface structure (similar to nitrogen inversion in tertiary amines), preventing the transfer of chirality to the interfacial layer and kernel of the gold nanocluster by uniformizing the staple motifs.

Herein, we report a 2,4-dimethylbenzenethiolate (2,4-DMBT)-containing $Au_{138}$ nanocluster with chiral configurations of the aryl groups on its surface but achiral arrangements of the $-S-Au-S-$ motifs and kernel atoms. The dynamic and interconvertible phenomenon in the chiral surface layer of $Au_{138}(SR)_{48}$ ($R = 2,4$-$Me_2C_6H_3$) is verified by variable-temperature nuclear magnetic resonance (NMR) experiments. The calculated circular dichroism (CD) spectra provide strong evidence that its intrinsic chirality originates from the chiral surface layer as opposed to the interfacial layer and the kernel. Despite the strong binding ability of thiolates to gold species[25], the steric hindrance provided by the 2,4-DMBT ligands partially shields the transitional layer, making $Au_{138}(SR)_{48}$ the thiolate-protected nanocluster with uncoordinated gold atoms in the outermost shell of the kernel.

In addition, near-spherical $Au_{138}(SR)_{48}$ with closed electronic shells shows both molecule- and metal-like characters. Although $Au_{144}(SR)_{60}$ clusters were discovered to have optical features and quantifiable electron dynamics that are characteristic of metals[26–30], it is still controversial whether this cluster is in a metallic state[31]. Nevertheless, $Au_{133}(SR)_{52}$ turns out to be a molecule-like gold nanocluster[22]. Consequently, identifying $Au_{138}(SR)_{48}$ as a transitional starting point from molecule-like gold nanoclusters to plasmonic nanoparticles is critical for shedding light on the mystery of plasmon excitation and understanding the mechanisms of localized surface plasmon resonance (LSPR)[32,33].

## Results

### Synthesis and structural analysis of $Au_{138}(SR)_{48}$

In our current study, 2,4-DMBT, a bulky and rigid aromatic thiolate ligand, was employed to prepare $Au_{138}(SR)_{48}$ via a two-step synthetic method (Supplementary Fig. 1)[34]. SCXRD analysis reveals that the cluster consisting of 138 gold atoms and 48 thiolates crystallizes in the cubic $Fm$-$3c$ space group. As shown in Fig. 1 and Supplementary Fig. 2, the $Au_{114}$ kernel with three concentric shells is protected by 24 monomeric [$-SR-Au-SR-$] staples. The average Au–Au bond lengths in the innermost Mackay icosahedral $Au_{12}$ shell and the second Mackay icosahedral $Au_{42}$ shell are 2.76 and 2.92 Å, respectively (Supplementary Fig. 3), which are roughly comparable to the experimental data of $Au_{144}(SCH_2Ph)_{60}$ ($Au_{12}$ shell: 2.75 Å; $Au_{42}$ shell: 2.91 Å)[23]. While theoretical[35] and experimental studies[23,36–39] suggested that the triple shell-by-shell $Au_{114}$ kernel tends to form thermally stable $Au_{144}(SR)_{60}$ nanoclusters, with all gold atoms in the third anti-Mackay icosahedral $Au_{60}$ shell coordinated by thiolates, only 48 gold atoms in the third shell of $Au_{138}(SR)_{48}$ bind to the [$-SR-Au-SR-$] staples, leaving the remaining 12 gold atoms uncoordinated (Fig. 1d). The average distance between the uncoordinated gold atom and the gold core atoms from the closest triangle of the second shell is 2.74 Å, about 0.1 Å shorter than the corresponding distance in the case of $Au_{144}(SCH_2Ph)_{60}$ (Supplementary Fig. 4), indicating that the uncoordinated gold atom is partially stabilized by the inner shell through stronger Au–Au bonding interactions. Because of the different coordination modes of the gold atoms in the transitional layer, the 12 pentagons in $Au_{138}(SR)_{48}$ include three distinct Au–Au bonds with lengths ranging from 2.83 to 3.43 Å, leading to a loss of all $C_5$ axes in the $Au_{114}$ kernel, one of the characteristic symmetry elements in icosahedral groups. In contrast to $Au_{144}(SCH_2Ph)_{60}$, which has 30 parallelograms in the third shell[23], $Au_{138}(SR)_{48}$ provides six rectangles on the (100) facets, as well as 24 quadrangles where one pair of opposite vertices are covered by the thiolates from the same staple motif (Fig. 1d). Each gold atom in the rectangle binds to one thiolate from four separate [$-SR-Au-SR-$] staples, with the

Au–Au bond lengths of 2.83 and 2.89 Å (Supplementary Fig. 5). Every three quadrangles form a subunit by sharing vertices with one another. These subunits, which contain an equilateral triangle in the center, are connected through the uncoordinated gold atoms (Fig. 1e).

To better understand the symmetry of $Au_{138}(SR)_{48}$, we simplify the substructure of the outermost $Au_{60}$ shell, which consists of two rectangles, one isosceles triangle, and one pentagon, as a directed line segment with its initial and terminal points located at the centers of the rectangles (Fig. 1f). This substructure is highly symmetrical, resulting in a mirror plane passing through the virtual line segment. Its direction is defined as being from the isosceles triangle to the pentagon. The 12 line segments form a regular octahedron with an equilateral triangle above each of its faces; each vertex of the octahedron serves as an initial point for two edges and a terminal point for the other two edges (Fig. 1g). As shown in Fig. 1h, i, the overall structure possesses three $C_2$ axes along the three-dimensional coordinates ($x$-, $y$-, and $z$-axes), as well as three horizontal mirror planes ($\sigma_h$) perpendicular to them. From the [111] direction of the simplified octahedron (Fig. 1j), one $S_6$ axis can be clearly identified; each face of the octahedron shares the same $C_3$ axis as the equilateral triangle above. Collectively, the $Au_{114}$ kernel has an unusual $T_h$ symmetry, with four $S_6$ axes along the centers of the opposite equilateral triangles in the third $Au_{60}$ shell.

### NMR investigations

While weak inter-cluster C–H···π interactions were identified by SCXRD (Supplementary Fig. 6), comprehensive NMR studies were conducted to investigate ligand–ligand bonding and thiolate distributions within the protective shell, both of which have a significant influence on the formation of the kernel structure. As shown in Fig. 2a, 48 2,4-DMBT ligands are evenly divided into four groups, with four sets of aromatic proton signals in a broad range of 8.87–4.30 ppm. All four 2,4-DMBT ligands in various local environments are found in every pentagonal $Au_5$ subunit of the third $Au_{60}$ shell (see the inset of Fig. 2a). The methyl groups at the 2-position of the thiolates with C–H···π interactions (marked in blue) are pointing toward each other, which provides substantial steric hindrance above the uncoordinated gold atom and precludes the potential binding of a fifth thiolate. The $^1$H-$^1$H COSY spectrum confirms that the exceptionally upshifted signals around 4.5 ppm are from the aromatic thiolates (Fig. 2b). When the C–H bond at the 5-position of a 2,4-DMBT is approached by another thiolate to exert C–H···π interactions, the corresponding hydrogen atom is greatly shielded by the delocalized conjugated π system. When two 2,4-DMBT ligands interact with each other via π–π stacking, the $^1$H NMR signals of the hydrogen atoms remaining within the other aromatic ring are significantly upshifted, whereas the deshielded hydrogen atoms residing outside the other ring give large chemical shifts. Furthermore, the signals indicating the correlations between the hydrogen atoms at the 5- and 6-positions of the same aromatic ring are very strong; the intensity of the signals resulting from the long-range couplings of the hydrogen atoms at the 3- and 5-positions is dramatically reduced (Fig. 2b). Taken together, all aromatic protons from the *cis* and *trans* [$-SR-Au-SR-$] staples, each three of which are joined together to form a separate protective unit A or B via C–H···π interactions, can be identified explicitly (Fig. 2c). Regarding the thiolate with both π–π stacking and C–H···π interactions in protective unit A, the chemical shift of its proton at the 5-position is largely upshifted, corresponding to 4.30 ppm. Due to the π–π stacking interaction, the proton at the 6-position gives a chemical shift of 6.32 ppm. With respect to the other thiolate from the same staple motif, which is strongly impacted only by π–π stacking, the protons at the 3- and 6-positions are shielded and deshielded, respectively, providing the largest chemical shift difference of 3.21 ppm. Since there is no π–π stacking within the

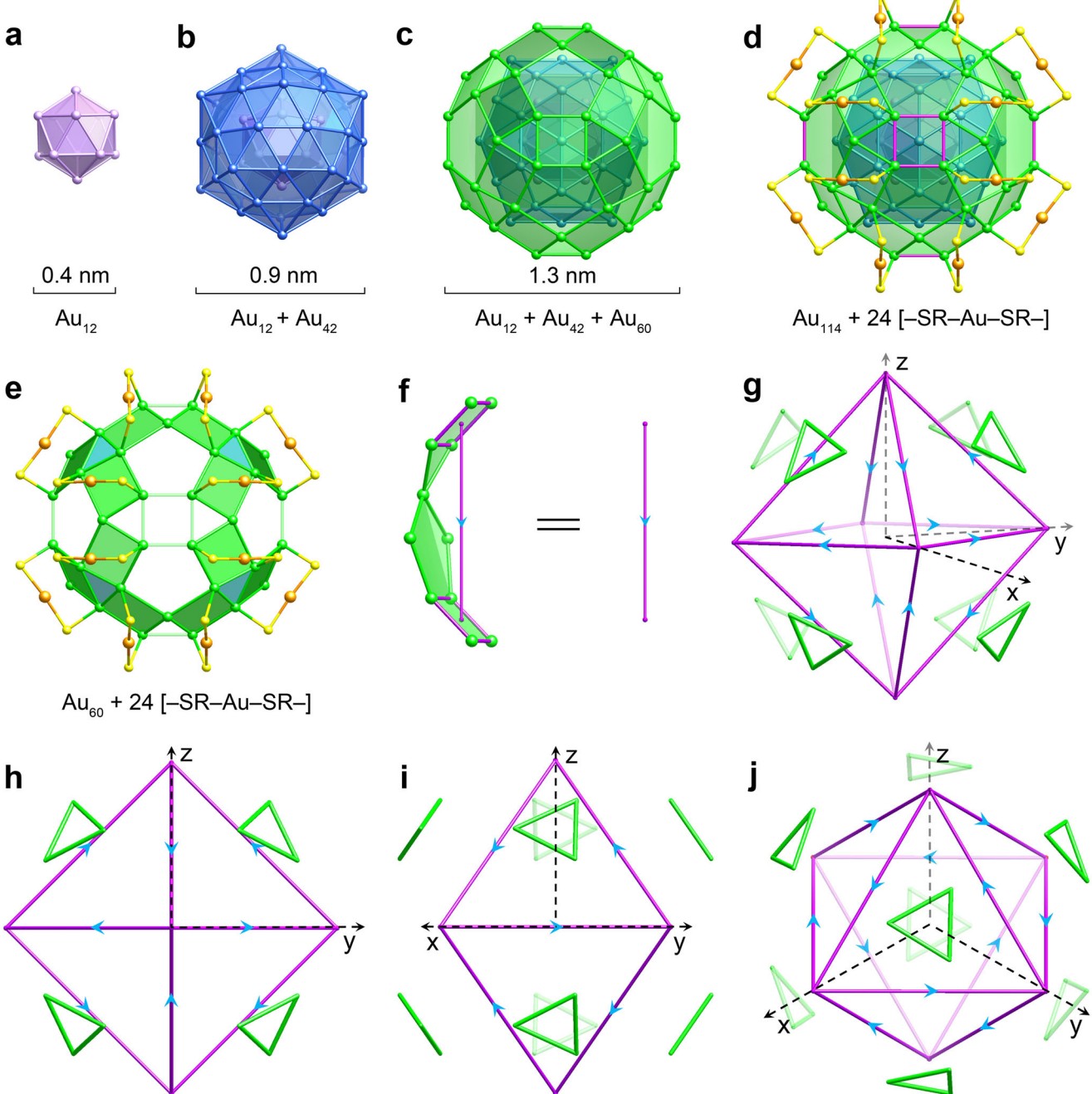

**Fig. 1 | X-ray crystallographic structure anatomy and symmetry analysis of Au₁₃₈(SR)₄₈.** **a** Innermost icosahedral $Au_{12}$ shell. **b** Second icosahedral $Au_{42}$ shell. **c** Third rhombicosidodecahedron-like $Au_{60}$ shell. **d** Total structure containing an $Au_{114}$ kernel and 24 [−SR−Au−SR−] staples. Rectangles are highlighted in magenta. **e** 24 quadrangles belonging to eight subunits produce eight equilateral triangles (marked in blue) in the third shell. **f** Substructure of the $Au_{60}$ shell with a plane of symmetry. **g** Connecting centers of the adjacent rectangles in the third shell produces a regular octahedron (marked in magenta), along with eight equilateral triangles (marked in green) above its faces. Views along the [100] (**h**), [110] (**i**), [111] (**j**) directions. Color labels in the chemical structures: Au, pink, blue, green, and orange; S, yellow. Aryl groups on thiolates are omitted for clarity.

*trans* [−SR−Au−SR−] staples of protective unit B, for the thiolate primarily with C−H···π interactions, the proton at the 5-position appears at 4.73 ppm, the second most upshifted signal in Au₁₃₈(SR)₄₈. This shielding effect is also observable in other nanocluster systems containing 2,4-DMBT ligand units assembled in a similar fashion[40]. For the thiolate lacking weak interactions with others, the chemical shift of the proton at the 5-position (6.90 ppm) is comparable to that of its counterpart in free 2,4-dimethylbenzenethiol (2,4-DMBTH) (Supplementary Fig. 7).

Although both protective units A and B are constructed by the C−H···π interactions between the three thiolates at their central locations, the aryl groups on these thiolates are arranged in anticlockwise and clockwise directions, respectively (Fig. 2c), imparting the intrinsic chirality to Au₁₃₈(SR)₄₈. To be more specific, in the other enantiomer, protective unit A', which is composed of three *cis* staple motifs, orients the aryl groups containing C−H···π interactions in a clockwise direction, whereas protective unit B', which is composed of three *trans* staple motifs, orients the corresponding aryl groups in an

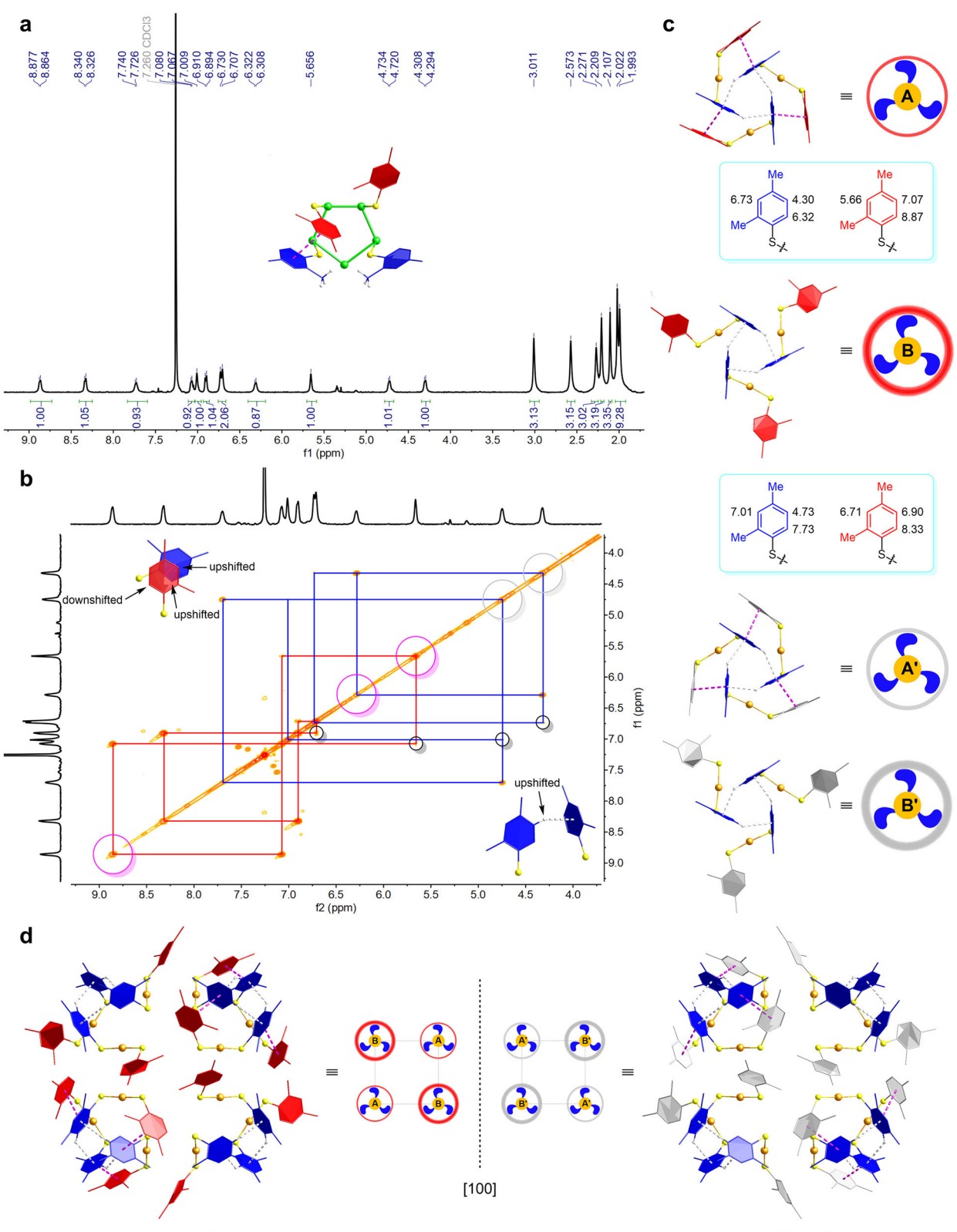

**Fig. 2 | Analysis of aromatic thiolates in the protective shell. a** ¹H NMR spectrum of Au₁₃₈(SR)₄₈ in chloroform-*d*. **b** ¹H–¹H COSY spectrum depicting the correlations between aromatic protons in four sets of 2,4-DMBT ligands. Pink and gray circles highlight protons significantly influenced by π–π stacking and C–H⋯π interactions, respectively. Long-range couplings between protons at the 3- and 5-positions of aromatic rings are demonstrated by black circles. **c** Schematic illustration of 12 [–SR–Au–SR–] staples in two separate protective units with assigned signals of all aromatic protons. **d** Enantiomeric distributions of aryl groups on the [–SR–Au–SR–] staples viewed from the [100] direction. The thin and thick circles with three fan blades inside depict the protective units with *cis* and *trans* staple motifs, respectively. The directions of the blue fan blades reflect the orientations of the aryl groups containing C–H⋯π interactions; the red and gray circles represent the aryl groups on the periphery of the protective units from two enantiomers of Au₁₃₈(SR)₄₈. Color labels in the chemical structures: Au, green and orange; S, yellow; C, red, blue, and gray; H, white.

anticlockwise direction. As observed from different views of the $Au_{138}$ nanocluster's interfacial layer (Supplementary Fig. 8), the distributions of −S−Au−S− motifs are highly symmetrical, resulting in three horizontal mirror planes perpendicular to the $C_2$ axes. Therefore, the chirality of $Au_{138}(SR)_{48}$ is solely determined by the enantiomeric distributions of the aryl groups in the thiolate assembly. Most significantly, $Au_{138}(SR)_{48}$ changes from one of its enantiomeric forms to the other when protective units A and B transform into protective units B′ and A′, respectively (Fig. 2d, Supplementary Figs. 9 and 10). In other words, the enantiomer interconversion can be achieved by simply altering the orientations of the aryl groups lacking C−H···π interactions (i.e., switching between the aryl groups marked in red and gray), while the arrangements of the aryl groups assembled via C−H···π interactions remain unchanged.

To account for the planes of symmetry in the interfacial layer (Fig. 1d and Supplementary Fig. 8), as well as the presence of eight identical equilateral triangles in the third $Au_{60}$ shell (Fig. 1e), we propose that the aryl groups on the periphery of the protective units could shuttle back and forth on the basis of a "pivot hinge", namely the highly symmetrical $[Au-S-Au-S-Au]_2$ moieties (Fig. 3a), thus uniformizing all the [−SR−Au−SR−] staples to protect and influence the kernel of $Au_{138}(SR)_{48}$. Regardless of whether the aromatic thiolates have π−π stacking, the Au−S bond lengths and the Au−S−Au bond angles are constant in every [Au−S−Au−S−Au] unit. However, as a result of the π−π stacking interactions, the Au−S−C bond angles are dramatically increased in *cis* staple motifs (Supplementary Fig. 11), indicating a higher degree of sp$^2$ character of the sulfur atom for configurational inversion[41]. As illustrated in Fig. 3b, when the aryl group on sp$^2$-hybridized $S^1$ starts to flip over along the Au−$S^1$−Au motif in Enantiomer 1, the π−π stacking interaction with the adjacent aryl group (marked in blue) is weakened. In the meantime, the C−$S^1$ single bond rotates clockwise to exert C−H···π interactions. During the relocation of the aryl group on $S^1$, the π−π stacking interaction between this aryl group and the one on $S^2$ promotes the conformational transformation from Intermediate 1 to Intermediate 2. The further clockwise rotation of the aryl group on $S^2$ produces Enantiomer 2 with a newly formed π−π stacking interaction. This highly dynamic intramolecular exchange process is strongly supported by the variable-temperature $^1$H-NMR studies (Fig. 3c). As the measurement temperature increased from 298 to 328 K, all the resonances from both aromatic and aliphatic regions were broadened substantially. Given that the chemical shift differences between the interchangeable protons are from 0.2 to 1.4 ppm, the rate of the interconversion between the two enantiomers of $Au_{138}(SR)_{48}$ was estimated[41,42] to be $10^1-10^2$ S$^{-1}$. The proton signals at the 6-position of the aryl groups with C−H···π interactions and the 3-position of the aryl groups without C−H···π interactions moved toward each other, indicating that the π−π stacking interactions were weakened on one side but strengthened on the other. According to the $^1$H−$^1$H COSY and HMBC spectra of $Au_{138}(SR)_{48}$ (Supplementary Figs. 12 and 14), the two sets of aliphatic proton signals that shifted closer together upon heating are from the methyl groups at the 2-position of the aromatic rings (Fig. 3c). The two methyl groups at down fields belong to the thiolates with C−H···π interactions, which are in close proximity to the uncoordinated gold atom on the kernel surface, while the other two with regular chemical shifts belong to the thiolates without C−H···π interactions. After cooling to 298 K, the $^1$H NMR spectrum of $Au_{138}(SR)_{48}$ returned to its original form (Supplementary Fig. 15), demonstrating the configurational reversibility and thermal stability of $Au_{138}(SR)_{48}$ in a chloroform-*d* solution.

## Studies on CD spectra
To provide further evidence that the chirality of $Au_{138}(SR)_{48}$ is solely a result of the arrangements of aromatic substituents in its surface layer, the optimized structures and the CD spectra of both simplified

$Au_{138}(SH)_{48}$ and $Au_{138}(SR)_{48}$ with full ligand sets were analyzed using density functional theory (DFT) computations (Fig. 4a). For both cluster models with a *T* symmetry, the CD spectra of different enantiomers are opposite in sign and equal in magnitude (Fig. 4b and Supplementary Fig. 16). As anticipated, the CD signal is significantly more pronounced when H is replaced by $2,4-Me_2C_6H_3$, demonstrating that the ligand substituents have a profound effect on the optical activity and chirality of $Au_{138}(SR)_{48}$. When investigating the $Au_{138}(SH)_{48}$ model, the H atoms can be placed in positions where the whole structure's $T_h$ symmetry is preserved. The calculated CD signal for this virtual cluster vanishes completely, indicating chirality loss in the inner layers of $Au_{138}(SR)_{48}$. Moreover, the Hausdorff chirality measure values[43] for the $Au_{114}$ kernel and the $(-S-Au-S-)_{24}$ interfacial layer were determined to be 0. These results strongly suggest that the chirality of $Au_{138}(SR)_{48}$ exists only in the surface layer, with the interfacial layer and the kernel remaining achiral.

## Origin of stability
The highly ordered geometric structures of the kernel and interfacial layer with a $T_h$ symmetry are one of the primary contributors to the exceptional stability of $Au_{138}(SR)_{48}$[44]. Additionally, the near-spherical $Au_{138}$ nanocluster has a valence electron count of 90 (that is, 138 − 48 = 90), which is one of the shell-closure magic numbers (i.e., 2, 8, 18, 58, 90, 92, 138,...) following the electronic cluster-shell model[45–47]. Its superatomic states, which are occupied by the 6s$^1$ electrons of the gold atoms, can be classified as $(1S^2|1P^6|1D^{10}|2S^21F^{14}|2P^61G^{18}|2D^{10}1H^{22})$. According to the calculated partial density of states (PDOS) curves for $Au_{138}(SR)_{48}$ with full ligand sets (Supplementary Fig. 18), the HOMO-LUMO gap was predicted to be 0.32 eV. The Au 5d and 6sp bands contribute the most to the low-lying occupied and unoccupied states, while the contributions from the thiolate ligands are somewhat smaller. The large Au 6sp atomic orbital components for the frontier levels suggest that these levels are indeed superatomic in nature, which is also consistent with the 90-electron count and the high symmetry ($T_h$) of the $Au_{114}$ kernel. The pictorial representations of the HOMO, LUMO, and other selected frontier molecular orbitals (MOs) are shown in Supplementary Fig. 19. In comparison, the major contributions from the organic substituents (C and H) start around −5 and −1 eV for the occupied and unoccupied energy levels, which originate from the π and π* orbitals of the aromatic rings, respectively.

## UV−vis absorption spectra
In the UV−vis region, $Au_{138}(SR)_{48}$ exhibited two major absorption peaks at 367 and 465 nm as well as three weak humps at 426, 489, and 521 nm (Fig. 5a). These excitonic signals reveal the discrete features of $Au_{138}(SR)_{48}$, and the broad shoulder at 521 nm suggests that $Au_{138}(SR)_{48}$ possesses a metal-like LSPR property[47]. The calculated UV-vis spectra and the major contributions to the spectral features are illustrated in Fig. 5, while a more detailed excited-state analysis and MO decomposition for the selected transitions are given in Supplementary Table 4. It is important to note that the calculated absorption spectra of Enantiomer 1 and Enantiomer 2 show no discernible differences. The theoretical spectra exhibit three major features around 360, 430, and 540 nm, which correspond well with the experimental data. As expected from the PDOS curves (Supplementary Fig. 18), the Au 5d → Au 6sp transitions contribute significantly to these characteristic features (Fig. 5b). For high-energy peaks 1 and 2, it is observed that the Au 6sp intraband contributions become relatively small, whereas the contributions from the Au 5d → π* ($2,4-Me_2C_6H_3$) and Au 6sp → π* ($2,4-Me_2C_6H_3$) transitions are increasingly coupled with the Au 5d → Au 6sp transitions, with an increase in excitation energies. These findings suggest that the interactions between the aromatic substituents of 2,4-DMBT ligands and the metal core have a substantial impact on the electronic and optical properties of the thiolate-protected gold

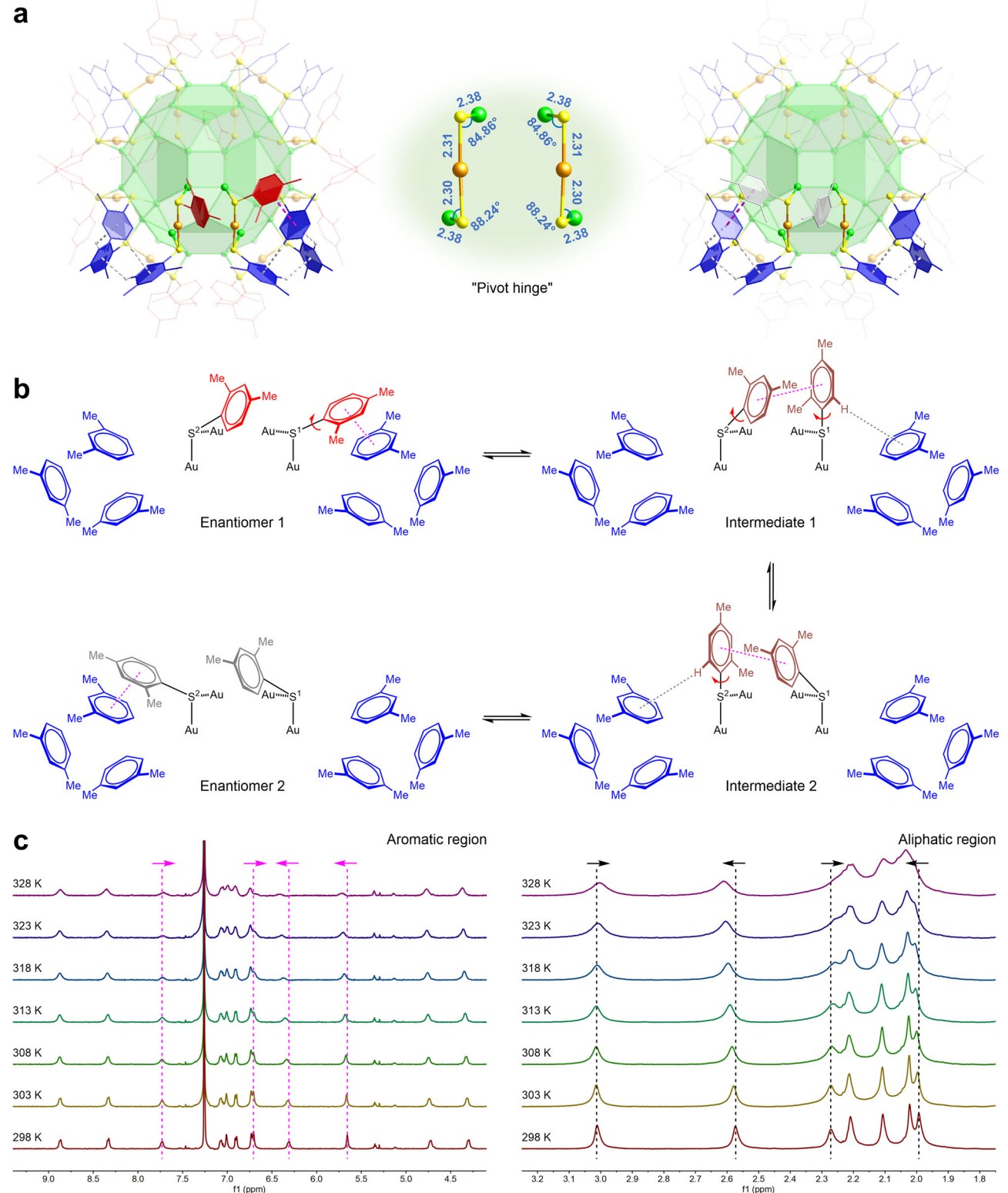

**Fig. 3 | An interconversion mechanism for structural analysis of $Au_{138}(SR)_{48}$. a** Highly symmetrical [Au−S−Au−S−Au]$_2$ moiety supporting the dynamic behaviors of aromatic rings. Color labels: Au, green and orange; S, yellow; C, red, blue, and gray; H, white. **b** A schematic representation of the interconversion process between two enantiomers of $Au_{138}(SR)_{48}$. **c** Variable-temperature $^1$H-NMR spectra covering both aromatic and aliphatic regions. Pink dashed lines highlight the migration of aromatic proton peaks influenced by π–π stacking interactions. Black dashed lines illustrate the configurational dynamics in methyl groups.

cluster[48,49]. For low-energy peak 3, however, there are more considerable contributions from Au $6sp$ intraband transitions in the frontier MOs, which are likely responsible for the metallic properties of $Au_{138}(SR)_{48}$.

## Compositional characterization

Electrospray ionization time-of-flight mass spectrometry (ESI−TOF−MS) in a positive-ion mode was chosen to investigate the molecular composition of the desired nanocluster. As shown in

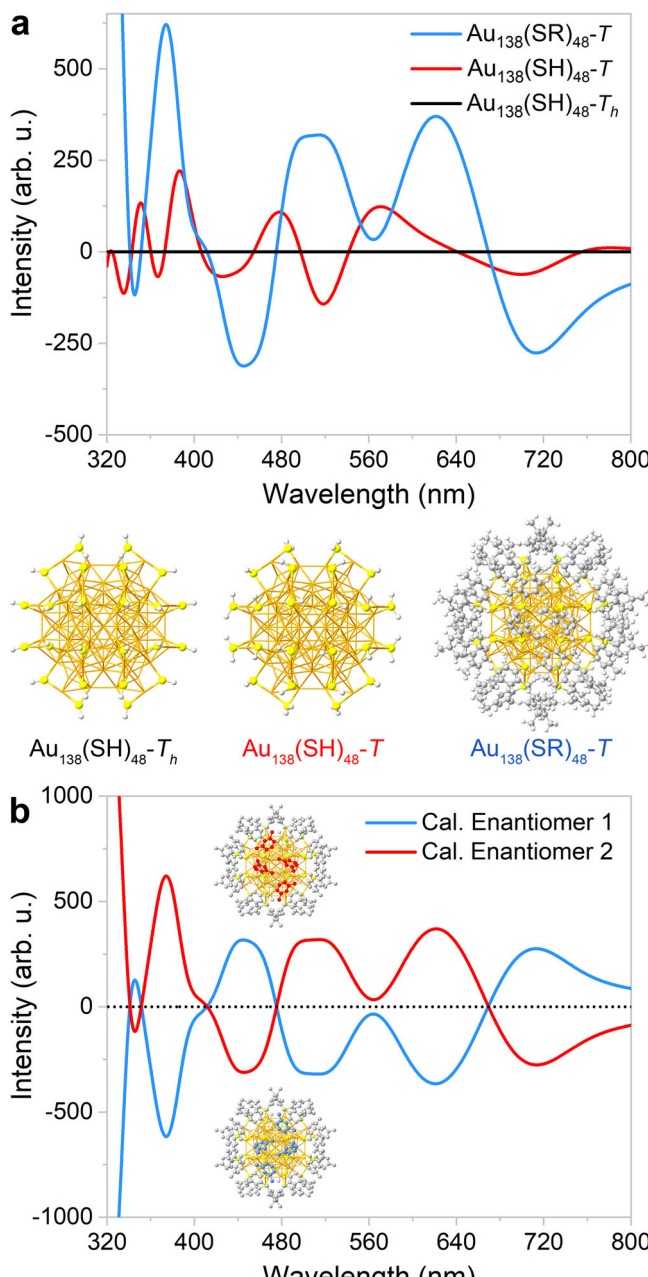

**Fig. 4 | Computed CD spectra for chirality determination. a** Comparing CD spectra of the related $Au_{138}$ clusters obtained by simplified time-dependent density functional theory (STDDFT) formalism and PBE/SVP level of theory. For the $Au_{138}(SH)_{48}$ cluster with a $T_h$ symmetry, H atoms were placed on preserving the symmetry of the remaining $Au_{138}S_{48}$ fragment. For the other two clusters, geometry optimizations were performed for the ligands without any constraints on symmetry. All the clusters were generated from the crystal structure of Enantiomer 2. **b** Mirror-image CD spectra of $Au_{138}(SR)_{48}$. The optimized enantiomeric structures are depicted in greater detail in Supplementary Fig. 17. Source data are provided as a Source Data file.

Supplementary Fig. 20, the peak at $m/z = 11256.89$ Da is associated with $[Au_{138}(2,4\text{-DMBT})_{48} + 3H]^{3+}$, and the 8442.87 Da signal corresponds to $[Au_{138}(2,4\text{-DMBT})_{48} + 4H]^{4+}$. The ESI−TOF−MS data demonstrate that $Au_{138}(SR)_{48}$ is a neutral molecule. According to the energy dispersive spectroscopy and mapping analysis, $Au_{138}(SR)_{48}$ primarily contains monodispersed Au and S elements in its crystals (Supplementary Fig. 21). The gold valence states of $Au_{138}(SR)_{48}$ were

determined using X-ray photoelectron spectroscopy (XPS) (Supplementary Fig. 22): the peaks at 84.23 and 87.97 eV were attributed to Au(0), while the peaks at 84.88 and 88.59 eV were attributed to Au(I)[50].

### Ultrafast electron dynamics

To elucidate the excited-state dynamics and molecular/metallic nature of $Au_{138}(SR)_{48}$, ultrafast transient absorption (TA) measurements were carried out upon excitation at 380 nm. According to the TA data map, $Au_{138}(SR)_{48}$ shows a positive excited-state absorption (ESA) at 485 nm and a negative ground-state bleaching (GSB) at 550 nm, respectively (Fig. 6a). The relaxation dynamics species associated spectra of $Au_{138}(SR)_{48}$ show three major components: a 4 ps bleach recovery component attributed to electron−phonon (e−p) relaxation, followed by a 76 ps component and a long-lived species indicating molecule-like nature of nanoclusters[47] (Fig. 6b). The TA spectra demonstrate that the ESA and GSB signals grow in the time windows from 0 to 1 ps and attenuate rapidly within 20 ps (Fig. 6c, d). The excitation-pulse-energy-dependent kinetics were investigated using TA measurements; the spectral features of $Au_{138}(SR)_{48}$ were essentially consistent at all pump fluences (Supplementary Fig. 23). As the pump energy increased, so did the contributions of the GSB and ESA components (Fig. 6e). The e−p relaxation time is highly sensitive to all pump power (Fig. 6f and Supplementary Fig. 24), confirming the metal-like character of $Au_{138}(SR)_{48}$[51]. The intrinsic e−p relaxation time ($\tau_0$) obtained via extrapolation is 1.58 ps for $Au_{138}(SR)_{48}$. The corresponding e−p coupling constant (G) was calculated to be $1.25 \times 10^{16}$ W m$^{-3}$ K$^{-1}$ (see Methods for details), which was similar to the reported value for $Au_{144}(SCH_2Ph)_{60}$ ($1.68 \times 10^{16}$ W m$^{-3}$ K$^{-1}$)[26]. The minor variations could be attributed to the different symmetries and valence electron counts of the $Au_{114}$ kernels as well as the different arrangements of the [−SR−Au−SR−] staple motifs. The UV−vis absorption spectra of $Au_{138}(SR)_{48}$ in toluene remained unchanged before and after the laser excitation (Supplementary Fig. 25), demonstrating the high stability of the cluster samples during the TA measurements.

### Discussion

We present here an example of achiral ligand-protected gold nanoclusters with a highly dynamic chiral surface. The facile interconversion between the two enantiomers renders the $Au_{114}$ kernel and the patterns of −S−Au−S− motifs in the interfacial layer a $T_h$ symmetry. Comprehensive NMR studies were employed to identify the π−π stacking and C−H⋯π interactions between the aromatic rings and to monitor the intramolecular exchange process in the protective shell at variable temperatures. The calculated CD spectra clearly indicate that only the chiral assembly of organic substituents contributes to intrinsic chirality. As supported by the U−vis absorption and TA data, $Au_{138}(SR)_{48}$ has both molecular and metallic nature, narrowing down the transitional size range for a gold nanocluster to evolve from a molecular to a metallic state. Further efforts to utilize uncoordinated gold atoms in the $Au_{138}$ nanocluster's kernel as intrinsic active sites for cluster transformations and catalysis are currently underway in our laboratory.

### Methods

#### Materials and reagents

All chemicals and solvents were commercially available and used without further purification. Tetrachloroauric(III) acid trihydrate (99% metal basis) was purchased from Aladdin Co., Ltd. Sodium borohydride (99.8%) was purchased from Sigma-Aldrich. Triphenylphosphine (>99%), 2,4-dimethylbenzenethiol (98%) were purchased from Saan Chemical Technology Co., Ltd. Methanol (>99%), dichloromethane (>99%), toluene (>99%), hexanes (>99%), and

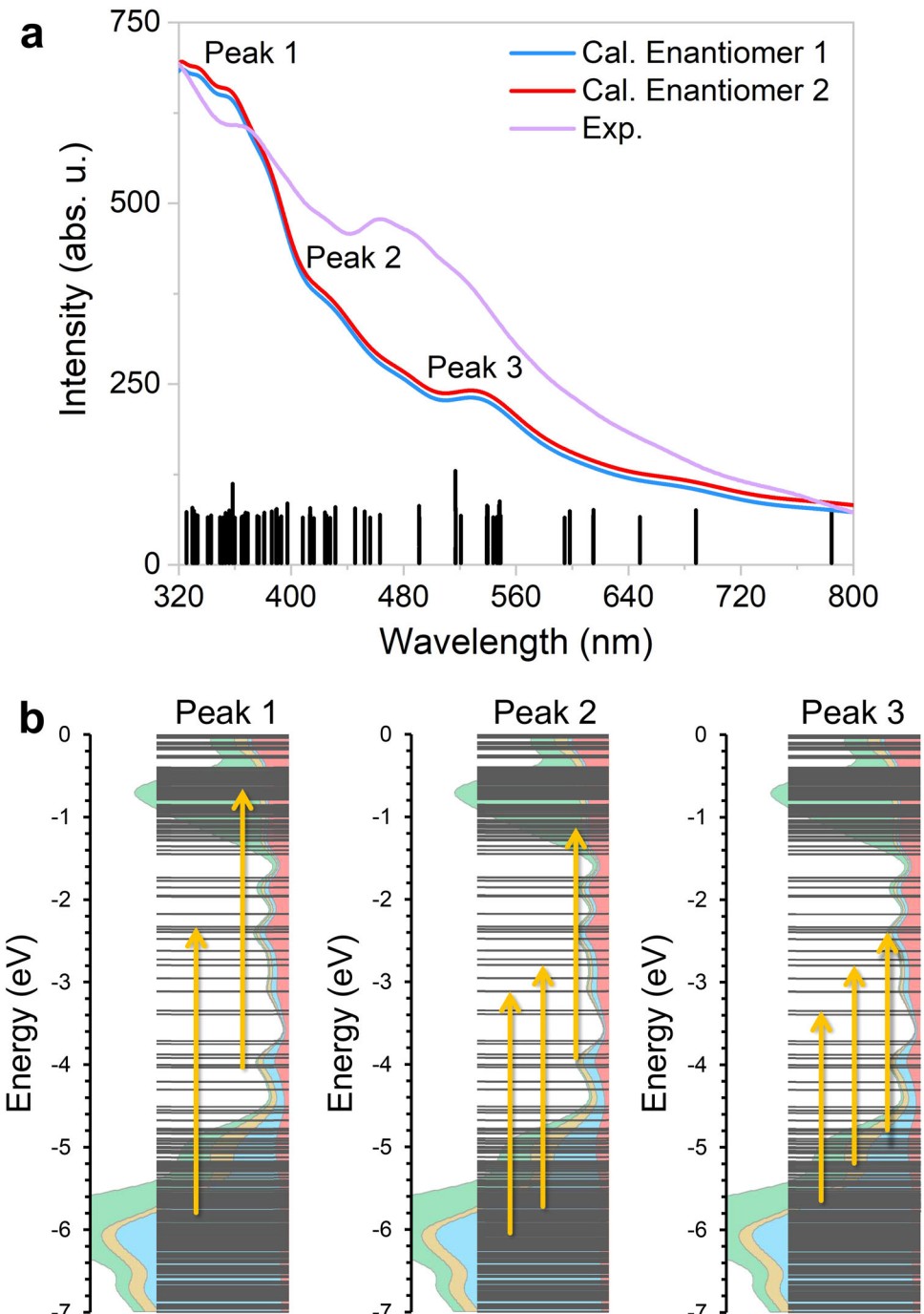

**Fig. 5 | Investigation of UV–vis absorption. a** UV–vis absorption spectrum (purple line) of Au$_{138}$(SR)$_{48}$ in dichloromethane at room temperature[48] and the calculated spectra (red and blue lines) of the two enantiomers with STDDFT formalism and PBE/SVP level of theory. The stick spectra are illustrated for the selected 100 transitions with the highest oscillator strength. **b** Excited-state analysis for the origins of peak 1 (≈360 nm), peak 2 (≈430 nm), and peak 3 (≈540 nm), respectively. Arrows represent the major contributions from occupied → virtual transitions for each peak. Supplementary Table 4 shows a more detailed orbital composition analysis along with transition information for the selected excited states. Source data are provided as a Source Data file.

sulfuric acid (98%) were purchased from Tianjin Kemiou Chemical Reagent Co., Ltd.

**Instrumentation**

The SCXRD data were collected with a Bruker D8 VENTURE CMOS PHOTON 100 diffractometer with a Helios MX multilayer monochromator using Cu Kα radiation ($\lambda = 1.54178$ Å). The structure was solved by direct methods using the SHELXT program[52] and refined by full-matrix least-squares on $F^2$ with anisotropic displacement parameters for all non-hydrogen atoms using the SHELXL program[53].

The ESI–TOF–MS data were recorded on a Waters Q-TOF mass spectrometer using a Z-spray source. The sample was prepared by dissolving the nanoclusters in dichloromethane (0.5 mg mL$^{-1}$). For the positive-ion mode detection, the sample was directly infused into the chamber at 5 mL min$^{-1}$. The source temperature was maintained at 70 °C, the spray voltage was 2.20 kV, and the cone voltage was adjusted to 60 V.

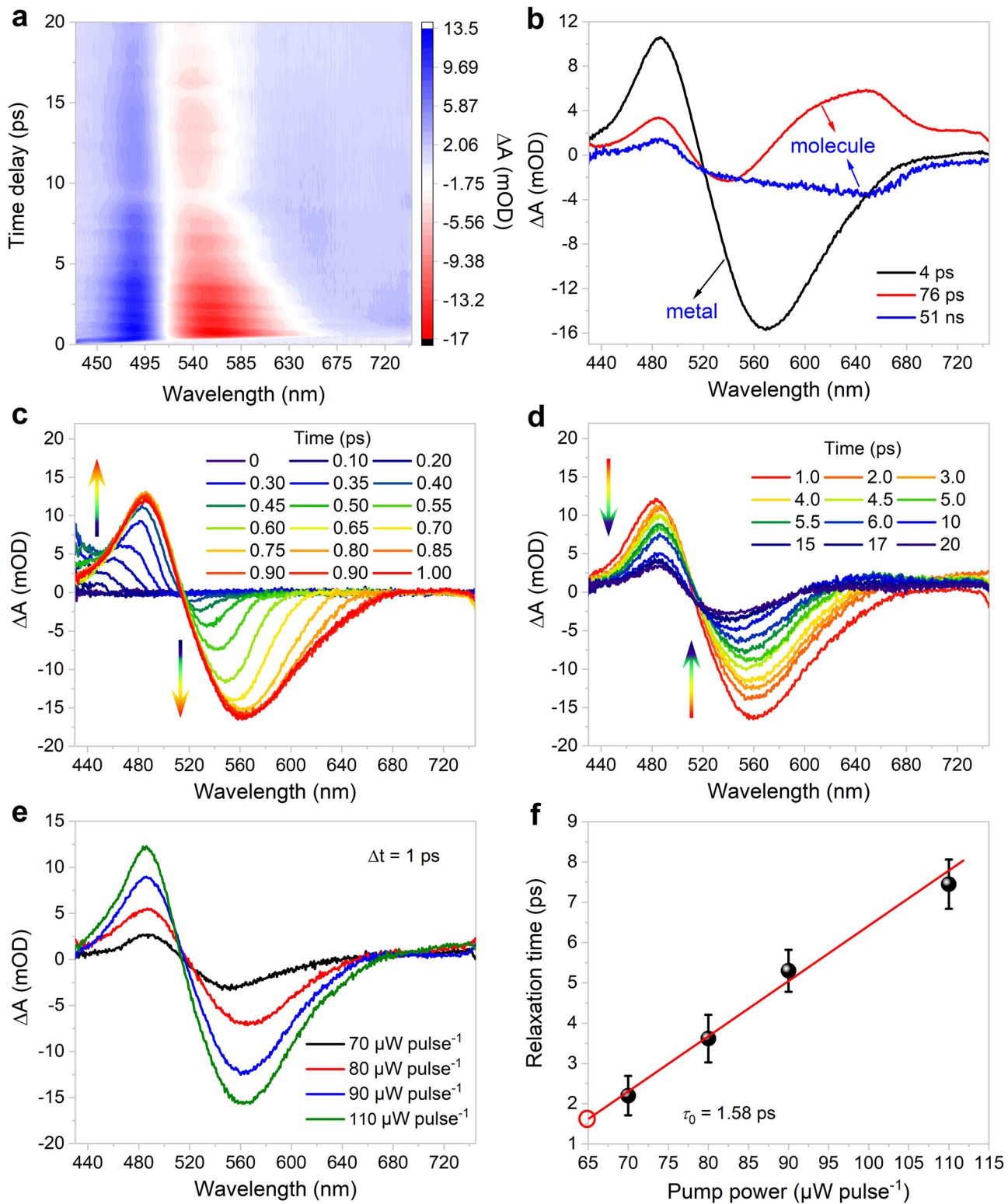

**Fig. 6 | Optical properties of Au₁₃₈(SR)₄₈.** **a** TA data map pumped at 380 nm in toluene at 110 μW pulse⁻¹. **b** Species-associated spectra (from global fit analysis). **c** TA spectra within 1 ps. **d** TA spectra as a function of time delay between 1 and 20 ps. **e** TA spectra measured with different pump energies. **f** The e–p relaxation time as a function of 380 nm pump fluence. The error bars represent the error range in the fitting results of multiexponential decay. Source data are provided as a Source Data file.

All UV–vis absorption spectra were acquired in the 200–800 nm range using a Cary3500 spectrophotometer (Agilent) at room temperature.

The elemental mapping data were collected on an EM-30 AX PLUS microscope (South Korea, COXEM company).

The XPS surface investigation was carried out on a Thermo ESCALAB 250Xi system, and the spectra were analyzed using the Thermo Scientific Avantage Data System software.

¹H, ¹³C NMR, ¹H–¹H COSY, and HMBC spectra were recorded on a Bruker 600 (600 MHz) or Bruker 500 (500 MHz) spectrometer in

chloroform-*d*. Chemical shifts were quoted in parts per million (ppm) referenced to 7.26 ppm of chloroform-*d* or the center line of a triplet at 77.0 ppm of chloroform-*d*.

## Synthetic procedure for Au$_{138}$(SR)$_{48}$

In a round bottom flask, 100 mg tetrachloroauric(III) acid trihydrate (0.254 mmol) and 50 mg triphenylphosphine (0.076 mmol) were initially dissolved in 20 mL dichloromethane/methanol (1:1 v/v). After 10 min slow stirring, 200 μL 2,4-DMBTH (1.45 mmol) was added, and a white solid gradually formed. Totally, 50 mg of freshly prepared sodium borohydride (1.3 mmol) dissolved in 3 mL ice-cold water was then introduced to the reaction mixture, followed by the addition of 1 mL sulfuric acid aqueous solution (40 wt%). The gold nanoclusters were allowed to grow for over 10 h. In the presence of excess methanol, the precursors containing multi-sized gold clusters were collected by centrifugation. In the second step, a two-phase reaction was involved. In detail, the obtained precursors and 200 μL 2,4-DMBTH (1.45 mmol) were dissolved in 10 mL toluene, and 10 mL water was used to separate the solution into two phases. The resulting solution was heated at 75 °C for 12 h. Black solids participated upon the addition of excess methanol. The residues were centrifuged and washed with methanol several times to remove free thiols and other byproducts. The crude product was purified using preparative thin-layer chromatography (PTLC) with dichloromethane/hexanes (1:2 v/v) as the eluent. Purified Au$_{138}$(SR)$_{48}$ was collected and extracted three times with dichloromethane. The combined organic layers were concentrated in a vacuum. Black block-like crystals of Au$_{138}$(SR)$_{48}$ suitable for single-crystal X-ray analysis were grown in dichloromethane/hexanes (1:1 v/v) at room temperature for 2 weeks. The brief synthesis route is displayed in Supplementary Fig. 1.

### $^1$H NMR (CDCl$_3$, 500 MHz)

δ 8.87 (d, *J* = 6.5 Hz, 12H), 8.33 (d, *J* = 7.0 Hz, 12H), 7.73 (d, *J* = 7.0 Hz, 12H), 7.07 (d, *J* = 6.5 Hz, 12H), 7.00 (s, 12H), 6.90 (d, *J* = 8.0 Hz, 12H), 6.73 (s, 12H), 6.71 (s, 12H), 6.32 (d, *J* = 7.0 Hz, 12H), 5.66 (s, 12H), 4.73 (d, *J* = 7.0 Hz, 12H), 4.30 (d, *J* = 7.0 Hz, 12H), 3.01 (s, 36H), 2.58 (s, 36H), 2.27 (s, 36H), 2.21 (s, 36H), 2.11 (s, 36H), 2.02–1.99 (m, 108H).

### $^{13}$C NMR (CDCl$_3$, 600 MHz)

δ 140.4, 137.6, 136.9, 136.44, 136.39, 136.2, 136.0, 135.7, 135.4, 135.1, 134.5, 134.2, 133.8, 130.8, 130.6, 129.91, 129.88, 129.8, 128.6, 127.6, 127.0, 126.7, 124.9, 29.3, 22.1, 21.5, 21.2, 20.7, 20.5 (multiple carbons overlapped).

### ESI-MS (*m/z*)

calcd. for C$_{384}$H$_{435}$S$_{48}$Au$_{138}$[M + 4H]$^{3+}$: 11256.88; found: 11256.89; calcd. for C$_{384}$H$_{436}$S$_{48}$Au$_{138}$ [M + 4H]$^{4+}$: 8442.86; found: 8442.87.

## Computational studies

All computations were performed with ORCA v5.03 program package[54] using PBE[55] functional and Def2-SVP[56] basis set. The Au$_{138}$(SR)$_{48}$ clusters with full ligand sets were mainly considered, but calculations of the simplified Au$_{138}$(SH)$_{48}$ model were also performed for comparison. To analyze the geometries, a constrained optimization was conducted in which R groups were allowed to relax while Au and S atoms were fixed in their experimentally determined positions in the crystal structure. Orbital decompositions, fragment, and angular-momentum resolved PDOS curves were obtained with the Multiwfn program[57]. In the case of PDOS curves, a Gaussian broadening of 0.2 eV FWHM was applied to discrete energy levels.

For the excited-state calculations, STDDFT formalism, as implemented in ORCA code, was employed to investigate the corresponding optical properties[58,59]. It should be noted that similar approximate TDDFT methods have shown excellent agreement with standard TDDFT formalism for ligand-protected Au and Ag clusters, as well as plasmonic systems[60–62]. In order to reduce the occupied → virtual

transition space, the Pthresh parameter, which controls the coupling threshold, was set to 10$^{-4}$. In our test calculations with a simplified Au$_{138}$(SH)$_{48}$ system, the introduction of this Pthresh had a minimal effect on the calculated spectra. For Au$_{138}$(SR)$_{48}$ with full ligand sets, ≈20000 excited states were calculated to cover the 0–4 eV energy range, whereas the calculation of ≈10000 excited states was sufficient in the case of Au$_{138}$(SH)$_{48}$ for the same energy range. Both CD and UV-vis spectra were generated with a Gaussian broadening of 0.2 eV FWHM.

## TA measurements

The data of electron dynamics were collected at room temperature using a Spectra-Physics Tsunami Ti:Sapphire (Coherent; 800 nm, 100 fs, 7 mJ pulse$^{-1}$, and 1 kHz repetition rate) as the laser source and a Helios spectrometer (Ultrafast Systems LLC). Briefly, the 800 nm output pulse from the regenerative amplifier was split in two parts. 95% of the output from the amplifier is used to pump a TOPAS optical parametric amplifier, which generates a wavelength-tunable laser pulse from 320 to 1600 nm as a pump beam in a Helios transient absorption setup (Ultrafast Systems Inc.). 380 nm pump beam was used for the measurements. The remaining 5% of the amplified output is focused onto a sapphire crystal to generate a white light continuum used for the probe beam in our measurements (320–780 nm). The pump beam was depolarized and chopped at 1 kHz, and both pump and probe beams were overlapped in the sample for magic angle transient measurements. Samples were vigorously stirred in all the measurements. The toluene solution for TA measurements was prepared from crystals of the gold nanocluster. No degradation of the sample was observed, as revealed from UV-vis absorption spectra before and after the measurements (Supplementary Fig. 25). Species-associated spectra were obtained by fitting the principal kinetics deduced from single value decomposition analysis. The growth of the kinetics was included in the instrument response. The room-temperature time constant obtained in this manner was then converted to G of the nanocluster using the equation[51]: $G = \gamma \times T_0/\tau_0$, where $\gamma = 66$ J m$^{-3}$ K$^{-2}$ for gold, $T_0$ is the room temperature, and $\tau_0$ is the intrinsic e–p relaxation time.

## Reporting summary

Further information on research design is available in the Nature Portfolio Reporting Summary linked to this article.

## Data availability

The data that support the findings of this study are available from the corresponding author upon request. Source data are provided in this paper. Crystallographic data for the structure of Au$_{138}$(SR)$_{48}$ have been deposited at the Cambridge Crystallographic Data Centre under deposition number CCDC 2195004. Copies of the data can be obtained free of charge via https://www.ccdc.cam.ac.uk/structures/. Source data are provided in this paper.

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

## Acknowledgements

The authors gratefully acknowledge The University of Hong Kong, the Research Grants Council of the Hong Kong Special Administrative Region, People's Republic of China (27301820, J.H., 17313922, J.H.), the Croucher Foundation, the Innovation and Technology Commission (HKSAR, China), and the National Natural Science Foundation of China (22201236, J.H., 32071421, X.L.) for their financial support. The DFT and STDDFT calculations for this project were performed at TUBITAK ULAK-BIM, High Performance and Grid Computing Center (TRUBA resources). The authors thank L.-L. Yan, for helpful discussions.

## Author contributions

L.-J.L. synthesized the gold nanoclusters and conducted most of the experiments for characterization with assistance from S.Z. F.A. carried out computational studies upon discussion with J.H. D.L. analyzed the data of transient absorption measurements, T.N. collected ESI-MS data, and J.G. performed NMR experiments. X.L. provided insights into cluster characterization. J.H. directed the project and wrote the manuscript with contributions from L.-J.L. and F.A.

## Competing interests

The authors declare no competing interests.
