## [Peer Review File · Nature Communications]

Atomically precise gold nanoclusters at the molecular-to-metallic transition with intrinsic chirality from surface layersReviewers' Comments:

Reviewer #1:

Remarks to the Author:

The authors report the crystal structure, and an analysis thereof, of a new gold cluster in the transition size range between "molecular" and "metallic" clusters. The work is certainly interesting and deserves publication. The paper is, in general, very well written. Nonetheless, before publication, a few things should be clarified.

The authors state that the cluster is non-chiral. If I understand their arguments correctly, that is based on the analysis of the orientations and positions of the ligand moieties.

1) how sensitive is the XRD in the detection of enantiomers? In the past, problems have occurred with the resolution of the enantiomers of, e.g., the Au₁₄₄L₆₀. Could the authors please comment on this? Possibly connected: In supplementary figure 5, what is the meaning of the two different colors for the Au atoms? Is this in any way referring to the different enantiomers?

2) I have some doubts about the qualification of the cluster by the authors as non-chiral. They clearly say that "the chirality of Au₁₃₈ is solely determined by the enantiomeric distributions of the aryl groups in the thiolate assembly." However, apart from the general arguments they present, there is no explicit study of the non-chirality of the gold core. Also in other clusters, it's the arrangement of the ligands that might induce some chirality in the otherwise achiral metal part, as in the case of Au₁₄₄ which is mentioned by the authors [cf., Acc. Chem. Res. 2019, 52, 34-43]. It would be good if the authors could add a qualitative measure of the chirality of the core to quantitatively distinguish the present cluster from the situation of Au₁₄₄. That would also add some information concerning the easy interconversion that they discuss.

3) A clear weak point of the paper are the calculations presented, which could actually provide much more information.

- what is meant by "the electronic structure was further optimized and evaluated using DFT calculations..." In the SI, it sounds like the structures are not relaxed ("optimized") after the reduction of the ligands.

- the reduction of the ligands seems to be rather drastic, in particular here for the structures where steric hindrance and the uncoordinated gold atoms are supposed to play a role. What is even more surprising, however, is the lack of analysis of the calculations. While there are two "Projected DOS" (Supp. fig. 17), these projections seem to refer to contributions of the different atoms, all angular-momentum contributions added? It is not really explained. In addition, I do not clearly understand the supplementary fig. 17: what is the difference between panels a and b? In panel b, are the values stacked or covering each other? What is the zero of energy? BTW, the fact that the colors are changed between panel a and b is not helping the reader at all.

In addition, the statement in the manuscript that "the third Au₆₀ shell and the interfacial layer contribute the most to the total density of states in the energy region of -10... -5 eV, indicative of strong interaction between the thiolate ligands and the kernel gold atoms." seems unclear. First, I do not see this in the supp. fig. 17 (the inner gold core Au₄₂ is also very present, in particular when considering the lower number of atoms compared to the Au₆₀); second, the better reference about the "strong interaction..." would be, in addition to 44, this one: [J. Phys. Chem. Lett. 2021, 12, 9262–9268].

It would be necessary there to provide the SAC projections in order to support the statements that are made in the text, because there, while the contributions from the SAC/jellium-like states may be correctly discussed based on the electron counting, no explicit demonstration has been done. Thus the

calculations are a bit useless in this otherwise nice paper.

- The spectra shown in supp. fig. 18 are likewise interesting but not clearly described. What is the temperature? In order to draw the conclusions that the authors mention (metallic...) it seems necessary to discuss this better. In the cluster of very similar size, Au₁₄₄, it has been shown that a large number of peaks appear when going to low temperatures (this is the better place for reference 44 in the manuscript). Therefore, to make any judgment based on the spectra alone about metallicity (or not), the reader needs to know the temperature at which the spectra were taken. In particular, it is here that the strong interaction of the aromatic ligands with the core, as described in the above-mentioned JPCLett, becomes important for a possible induction of a plasmonic response by the increased electron delocalization, as it will certainly carry over to the present structure.

- mention of Ref. 43 for the electronic shell structures -- this doesn't seem to be an appropriate reference for it, although of course, there's a lot of discussion in the article about it. There is the Walter et al. PNAS 2010, and there are other earlier works (refs in that ref. 43...)

Apart from this, I find Fig. 1 extremely hard to interpret -- as a schematic image to show the mechanism it's difficult to "read," the authors might want to improve it.

As a side remark, supp. figs 8 and 9 are rather dark and difficult to see on my print-out.

Finally, I have not been able to obtain the structure that the authors said was submitted to <https://www.ccdc.cam.ac.uk/structures/>. I would appreciate receiving it if there is a second round of refereeing.

Reviewer #2:

Remarks to the Author:

The manuscript from Li-Juan Liu et al. is a nice experimental contribution in the field of chirality in nanosystems, in particular in chiral atomically precise nanoclusters. The message from this work is simple and bold: Au₁₃₈ cluster protected by 2,4 DMBT displays a chiral pattern of the external aromatic substituents which is NOT transferred to the inner metallic core. This is somehow unexpected, since usually the presence of an external chiral ligand arrangement (for example in the staple units) is able to induce a geometric rearrangement of the metallic core so that finally a chiral metallic kernel is obtained. A possible explanation of such observation is suggested by the authors, and is ascribed by fast dynamics of the aromatic rings. Moreover the molecular and metallic behaviours have been both observed for this system. The anatomy of the Au₁₃₈ cluster is very well explained, on the basis of X-ray diffraction, so the structural results are convincing. Analysis of the aromatic thiolates is also performed by means of NMR spectra. In summary, the experimental data have been obtained with appropriate techniques so the conclusions are convincing and robust. The conclusions are also general, at the moment there is increasing interest for chiral clusters, so the paper is of great potential interest for a quite wide scientific audience. The only weakness of the paper refers to the computational contribution. The DFT calculations have been performed substituting the 2,4 DMBT ligand with a SH group. This is a quite crude approximation, which would have been maybe acceptable for alkyl thiolates, but is not adequate for aryl thiolates. In fact aryl thiolates are 'coupled' by conjugation with plasmonic resonances and therefore may have dramatic impact on the optical response of the cluster. Moreover in the present work there was no attempt to rationalize (typically by TDDFT calculations) the photoabsorption spectrum (Fig. S18) which could reveal important information regarding the molecular/metallic behaviour. I recommend publication, but after the authors have improved DFT calculations and included a TDDFT analysis of the spectrum, which is rather common in studies like the present one.

Reviewer #3:

Remarks to the Author:

The authors present a new cluster, Au₁₃₈, determine its structure, study the dynamics of the ligand shell by NMR spectroscopy and investigate its photophysics. Although it is nice work I am wondering if the novelty is sufficient for a top journal like Nature Communications.

The authors mention three points why this work is of interest:

- 1) "... the first gold nanocluster whose inner structures are not asymmetrically induced by chiral patterns of the outermost aromatic substituents."
- 2) First nanocluster with uncoordinated surface gold atoms.
- 3) the smallest-sized gold nanocluster having both molecular and metallic properties.

Concerning point 1): The surface structure of the cluster is similar to the one of Ag₂₅(DMBT)₁₈ (the ligand is the same). Also, this cluster has a symmetric core and achiral arrangement of ligands. The NMR presented in the current paper resembles strongly the one in JPCP 2021, 125, 2524-2530.

Concerning point 2): This may be but how can the authors be sure...did they carefully check all the available structures?

Concerning point 3): This transition is not only a function of size but also of structure. If the transition has been assigned before to about 144 gold atoms, the re-assignment to 138 atoms seems not too exciting to this reviewer.

The authors write "...it was realized that the NCs' chirality might also be associated with the staple distributions or kernel structures, rather than solely being a result of chirality transfer from the external chiral organic components."

In my opinion it is not clear if chirality is rather transferred from outside inwards or from inside outwards as a general rule. (From inside outwards sees also possible, see Dolamic, Nature Commun. 7117 (2015))

We sincerely appreciate the reviewers' time and efforts in helping us improve the quality of our work. We conducted additional experiments and revised the manuscript in response to their constructive feedback and suggestions.

Reviewer #1

(1) how sensitive is the XRD in the detection of enantiomers? In the past, problems have occurred with the resolution of the enantiomers of, e.g., the Au₁₄₄L₆₀. Could the authors please comment on this? Possibly connected: In supplementary figure 5, what is the meaning of the two different colors for the Au atoms? Is this in any way referring to the different enantiomers?

Response: We thank the reviewer for the questions. Similar to the cases of Au₁₄₄(SR)₆₀, it is difficult to determine the absolute configurations of Au₁₃₈ using XRD measurements. Given that the enantiomers are interconvertible by switching the aryl groups in red and gray (see Fig. 4), the XRD data reveal that the site occupancy factor of the aryl rings in blue is 1, whereas the site occupancy factor of the aryl rings in red and gray is 0.5.

Pink and blue color in Supplementary Fig. 5 represent the clusters in two different layers along the [010] direction, rather than the different enantiomers. We have updated this figure and added the detailed description as stated below:

Supplementary Figure 5. Packing structure of Au_{138} in single crystals. Views from the [100] (a), [010] (b), and [111] (c) directions. Inset: inter-cluster C-H... π interactions between the protective shells. Color labels: Au, pink, blue (indicating two different layers viewed along the [010] direction); S, yellow; C, grey; H white.

(2) I have some doubts about the qualification of the cluster by the authors as non-chiral. They clearly say that "the chirality of Au138 is solely determined by the enantiomeric distributions of the aryl groups in the thiolate assembly." However, apart from the general arguments they present, there is no explicit study of the non-chirality of the gold core. Also in other clusters, it's the arrangement of the ligands that might induce some chirality in the otherwise achiral metal part, as in the case of Au144 which is mentioned by the authors [cf., Acc. Chem. Res. 2019, 52, 34-43]. It would be good if the authors could add a qualitative measure of the chirality of the core to quantitatively distinguish the present cluster from the situation of Au144. That would also add some information concerning the easy interconversion that they discuss.

Response: We thank the reviewer for the constructive comments. To distinguish **Au138** from **Au144** quantitatively, we computed the CD spectra of a series of $\text{Au}_{138}(\text{SR})_{48}$ ($\text{R} = \text{H}$ or $2,4\text{-Me}_2\text{C}_6\text{H}_3$) and the Hausdorff chirality measure values of the core structures. We have added a new paragraph and figure to the manuscript to further elaborate on the chirality:

“To provide further evidence that the chirality of **Au138** is solely a result of the arrangements of aromatic substituents in its surface layer, the optimized structures and the CD spectra of both simplified $\text{Au}_{138}(\text{SH})_{48}$ and **Au138** with full ligand sets were analyzed using density functional theory (DFT) computations (Fig. 5a). For both cluster models with a T symmetry, the CD spectra of different enantiomers are opposite in sign and equal in magnitude (Fig. 5b and Supplementary Fig. 15). As anticipated, the CD signal is significantly more pronounced when $\text{R} = 2,4\text{-Me}_2\text{C}_6\text{H}_3$ rather than H, demonstrating that the ligand substituents have a profound effect on the optical activity and chirality of **Au138**. When investigating the $\text{Au}_{138}(\text{SH})_{48}$ model, the H atoms can be placed in positions where the whole structure's T_h symmetry is preserved. The calculated CD signal for this virtual cluster vanishes completely, indicating chirality loss in the inner layers of **Au138**. Moreover, the Hausdorff chirality measure values⁴² for the Au_{114} kernel and the $(\text{-S-Au-S-})_{24}$ interfacial layer were determined to be 0. These results strongly suggest that chirality of **Au138** exists only in the surface layer, with the interfacial layer and the kernel remaining achiral.”

Fig. 5 | Computed CD spectra for chirality determination. a Comparing the CD spectra of **Au₁₃₈**-related clusters obtained by simplified time-dependent density functional theory (STDDFT) formalism and PBE/SVP level of theory. R = 2,4-Me₂C₆H₃. For the Au₁₃₈(SH)₄₈ NC with a *T_h* symmetry, H atoms were placed to preserve the symmetry of the remaining Au₁₃₈S₄₈ fragment. For the other two NCs, geometry optimizations were performed for the ligands without any constraints on symmetry. All the clusters were generated from the crystal structure of **Enantiomer 2**. **b** Mirror-image CD spectra of the two enantiomers of **Au₁₃₈**.

(3) A clear weak point of the paper are the calculations presented, which could actually provide much more information.

- what is meant by "the electronic structure was further optimized and evaluated using DFT calculations..." In the SI, it sounds like the structures are not relaxed ("optimized") after the reduction of the ligands.

- the reduction of the ligands seems to be rather drastic, in particular here for the structures where steric hindrance and the uncoordinated gold atoms are supposed to play a role. What is even more surprising, however, is the lack of analysis of the calculations. While there are two "Projected DOS" (Supp. fig. 17), these projections seem to refer to contributions of the different atoms, all angular-momentum contributions added? It is not really explained. In addition, I do not clearly understand the supplementary fig. 17: what is the difference between panels a and b? In panel b, are the values stacked or covering each other? What is the zero of energy? BTW, the fact that the colors are changed between panel a and b is not helping the reader at all.

Response: We thank the reviewer for the constructive comments. We re-calculated the structures of **Au₁₃₈** without the reduction of the ligands to H atoms and added the computational study section to Methods:

“All computations were performed with ORCA v5.03 program package⁵¹ using PBE⁵² functional and Def2-SVP⁵³ basis set. The Au₁₃₈(SR)₄₈ clusters with full ligand sets (R = 2,4-Me₂C₆H₃) were mainly considered, but calculations of the simplified Au₁₃₈(SH)₄₈ model were also performed for comparison. To analyze the geometries, a constrained optimization was conducted in which R groups were allowed to relax while Au and S atoms were fixed in their experimentally determined positions in the crystal structure. Orbital decompositions, fragment and angular-momentum resolved PDOS curves were obtained with Multiwfn program⁵⁴. In the case of PDOS curves, a Gaussian broadening of 0.2 eV FWHM was applied to discrete energy levels.”

After performing the new DFT calculations, we added the revised analysis part to the manuscript:

“According to the calculated partial density of states (PDOS) curves for **Au₁₃₈** with full ligand sets (Supplementary Fig. 16), the HOMO-LUMO gap was predicted to be 0.32 eV. The Au 5d and 6sp bands contribute the most to the low-lying occupied and unoccupied states, while the contributions from the thiolate ligands are somewhat smaller. The large Au 6sp atomic orbital components for the frontier levels suggest that these levels are indeed superatomic in nature, which is also consistent with the 90-electron count and the high symmetry (T_h) of the Au₁₁₄ kernel. The pictorial representations of the HOMO, LUMO, and other selected frontier molecular orbitals (MOs) are shown in Supplementary Fig. 17. In comparison, the major contributions from the organic substituents (C and H) start around -5 and -1 eV for the occupied and unoccupied energy levels, which originate from the π and π^* orbitals of the aromatic rings, respectively.”

Supplementary Figs. 16 and 17 have been updated based on the reviewer’s comments and the structure with full ligand sets:

Supplementary Figure 16. Calculated partial and total DOS curves for Au_{138} obtained by PBE/SVP level of theory. For Au atoms, contributions from sp (red) and d (blue) bands are shown separately. **a** PDOS curves for the energy range between -15 and 4 eV. **b** The same curves for a smaller energy range for clarity with a focus on the contributions to frontier levels.

Supplementary Figure 17. Energies and pictorial representations of occupied (**a**) and unoccupied (**b**) frontier molecular orbitals of **Au₁₃₈** obtained by PBE/SVP level of theory.

(4) In addition, the statement in the manuscript that "the third Au₆₀ shell and the interfacial layer contribute the most to the total density of states in the energy region of -10... -5 eV, indicative of strong interaction between the thiolate ligands and the kernel gold atoms." seems unclear. First, I do not see this in the supp. fig. 17 (the inner gold core Au₄₂ is also very present, in particular when considering the lower number of atoms compared to the Au₆₀); second, the better reference about the "strong interaction..." would be, in addition to 44, this one: [J. Phys. Chem. Lett. 2021, 12, 9262–9268].

It would be necessary there to provide the SAC projections in order to support the statements that are made in the text, because there, while the contributions from the SAC/jellium-like states may be correctly discussed based on the electron counting, no explicit demonstration has been done. Thus the calculations are a bit useless in this otherwise nice paper.

Response: We thank the reviewer for the constructive comments. We re-wrote the discussion of molecular orbitals based on the new DFT calculation data obtained with full ligand sets. The reference pointed out by the reviewer has been added to the manuscript. Please refer to the following paragraphs:

“According to the calculated partial density of states (PDOS) curves for **Au₁₃₈** with full ligand sets (Supplementary Fig. 16), the HOMO-LUMO gap was predicted to be 0.32 eV. The Au 5d and 6sp bands contribute the most to the low-lying occupied and unoccupied states, while the contributions from the thiolate ligands are somewhat smaller. The large Au 6sp atomic orbital components for

the frontier levels suggest that these levels are indeed superatomic in nature, which is also consistent with the 90-electron count and the high symmetry (T_h) of the Au₁₁₄ kernel. The pictorial representations of the HOMO, LUMO, and other selected frontier molecular orbitals (MOs) are shown in Supplementary Fig. 17. In comparison, the major contributions from the organic substituents (C and H) start around -5 and -1 eV for the occupied and unoccupied energy levels, which originate from the π and π^* orbitals of the aromatic rings, respectively.”

“As expected from the PDOS curves (Supplementary Fig. 16), the Au 5d \rightarrow Au 6sp transitions contribute significantly to these characteristic features (Fig. 6b). For high-energy peaks 1 and 2, it is observed that the Au 6sp intraband contributions become relatively small, whereas the contributions from the Au 5d $\rightarrow \pi^*$ (2,4-Me₂C₆H₃) and Au 6sp $\rightarrow \pi^*$ (2,4-Me₂C₆H₃) transitions are increasingly coupled with the Au 5d \rightarrow Au 6sp transitions, with an increase in excitation energies. These findings suggest that the interactions between the aromatic substituents of 2,4-DMBT ligands and the metal core have a substantial impact on the electronic and optical properties of the thiolate-protected gold cluster^{47,48}. For low-energy peak 3, however, there are more considerable contributions from Au 6sp intraband transitions in the frontier MOs, which are likely responsible for the metallic properties of **Au₁₃₈**.”

(5) - The spectra shown in supp. fig. 18 are likewise interesting but not clearly described. What is the temperature? In order to draw the conclusions that the authors mention (metallic...) it seems necessary to discuss this better. In the cluster of very similar size, Au₁₄₄, it has been shown that a large number of peaks appear when going to low temperatures (this is the better place for reference 44 in the manuscript). Therefore, to make any judgment based on the spectra alone about metallicity (or not), the reader needs to know the temperature at which the spectra were taken. In particular, it is here that the strong interaction of the aromatic ligands with the core, as described in the above-mentioned JPCLet, becomes important for a possible induction of a plasmonic response by the increased electron delocalization, as it will certainly carry over to the present structure.

Response: We thank the reviewer for the constructive comments. All the optical measurements were performed at room temperature; we have added the measuring temperature to the revised manuscript and SI. In addition, we included calculated UV-vis spectra to discuss inter- and intraband transitions as well as interactions between the metal core and aromatic ligands:

“The calculated UV-vis spectra and the major contributions to the spectral features are illustrated in Fig. 6, while a more detailed excited-state analysis and MO decomposition for the selected transitions are given in Table S4. It is important to note that the calculated absorption spectra of **Enantiomer 1** and **Enantiomer 2** show no discernible differences. The theoretical spectra exhibit three major features around 360, 430, and 540 nm, which correspond well with the experimental data. As expected from the PDOS curves (Supplementary Fig. 16), the Au 5d \rightarrow Au 6sp transitions contribute significantly to these characteristic features (Fig. 6b). For high-energy peaks 1 and 2, it is observed that the Au 6sp intraband contributions become relatively small, whereas the contributions from the Au 5d $\rightarrow \pi^*$ (2,4-Me₂C₆H₃) and Au 6sp $\rightarrow \pi^*$ (2,4-Me₂C₆H₃) transitions are increasingly coupled with the Au 5d \rightarrow Au 6sp transitions, with an increase in excitation

energies. These findings suggest that the interactions between the aromatic substituents of 2,4-DMBT ligands and the metal core have a substantial impact on the electronic and optical properties of the thiolate-protected gold cluster^{47,48}. For low-energy peak 3, however, there are more considerable contributions from Au 6sp intraband transitions in the frontier MOs, which are likely responsible for the metallic properties of **Au₁₃₈**.”

(6) - mention of Ref. 43 for the electronic shell structures -- this doesn't seem to be an appropriate reference for it, although of course, there's a lot of discussion in the article about it. There is the Walter et al. PNAS 2010, and there are other earlier works (refs in that ref. 43...)

Response: We appreciate the helpful comment from the reviewer. The following references have been selected to replace the previous Ref. 43:

44. Knight, W. D. *et al.* Electronic shell structure and abundances of sodium clusters. *Phys. Rev. Lett.* **52**, 2141–2143 (1984).
45. Walter, M. *et al.* A unified view of ligand-protected gold clusters as superatom complexes. *Proc. Natl. Acad. Sci. USA* **105**, 9157–9162 (2008).
46. Sakthivel, N. A. *et al.* The missing link: Au₁₉₁(SPh-^tBu)₆₆ Janus nanoparticle with molecular and bulk-metal-like properties. *J. Am. Chem. Soc.* **142**, 15799–15814 (2020).

(7) Apart from this, I find Fig. 1 extremely hard to interpret -- as a schematic image to show the mechanism it's difficult to "read," the authors might want to improve it.

Response: We appreciate the helpful comment from the reviewer. We have significantly modified the staples and kernel in each structure of Fig. 1 to distinguish the nanoclusters with and without the “outside-in” influence.

Fig. 1 | Intrinsic chirality in atomically precise gold NCs protected by achiral thiolate ligands.

(8) As a side remark, supp. figs 8 and 9 are rather dark and difficult to see on my print-out.

Response: We appreciate the helpful comment from the reviewer. Supplementary Figs. 8 and 9 now have a white background.

(9) Finally, I have not been able to obtain the structure that the authors said was submitted to <https://www.ccdc.cam.ac.uk/structures/>. I would appreciate receiving it if there is a second round of refereeing.

Response: We have deposited the crystal structure via the joint CCDC/FIZ Karlsruhe deposition service (Deposition Number 2195004). The data will be available after publication. We included the cif file in this revision to make it easier for the reviewer to access the data.

Reviewer #2

(1) The conclusions are also general, at the moment there is increasing interest for chiral clusters, so the paper is of great potential interest for a quite wide scientific audience. The only weakness of the paper refers to the computational contribution. The DFT calculations have been performed substituting the 2,4 DMBT ligand with a SH group. This is a quite crude approximation, which would have been maybe acceptable for alkyl thiolates, but is not adequate for aryl thiolates. In fact aryl thiolates are ‘coupled’ by conjugation with plasmonic resonances and therefore may have dramatic impact on the optical response of the cluster.

Response: We thank the reviewer for the constructive suggestion. We re-calculated the structures of Au_{138} using full ligand sets and added the computational study section to Methods:

“All computations were performed with ORCA v5.03 program package⁵¹ using PBE⁵² functional and Def2-SVP⁵³ basis set. The $\text{Au}_{138}(\text{SR})_{48}$ clusters with full ligand sets ($\text{R} = 2,4\text{-Me}_2\text{C}_6\text{H}_3$) were mainly considered, but calculations of the simplified $\text{Au}_{138}(\text{SH})_{48}$ model were also performed for comparison. To analyze the geometries, a constrained optimization was conducted in which R groups were allowed to relax while Au and S atoms were fixed in their experimentally determined positions in the crystal structure. Orbital decompositions, fragment and angular-momentum resolved PDOS curves were obtained with Multiwfn program⁵⁴. In the case of PDOS curves, a Gaussian broadening of 0.2 eV FWHM was applied to discrete energy levels.”

(2) Moreover in the present work there was no attempt to rationalize (typically by TDDFT calculations) the photoabsorption spectrum (Fig. S18) which could reveal important information regarding the molecular/metallic behaviour. I recommend publication, but after the authors have improved DFT calculations and included a TDDFT analysis of the spectrum, which is rather common in studies like the present one.

Response: We thank the reviewer for the constructive comments. We have performed the TDDFT calculations to obtain the computed spectra of both enantiomers and included a detailed analysis in the manuscript:

“The calculated UV-vis spectra and the major contributions to the spectral features are illustrated in Fig. 6, while a more detailed excited-state analysis and MO decomposition for the selected transitions are given in Table S4. It is important to note that the calculated absorption spectra of **Enantiomer 1** and **Enantiomer 2** show no discernible differences. The theoretical spectra exhibit three major features around 360, 430, and 540 nm, which correspond well with the experimental data. As expected from the PDOS curves (Supplementary Fig. 16), the Au 5d \rightarrow Au 6sp transitions contribute significantly to these characteristic features (Fig. 6b). For high-energy peaks 1 and 2, it is observed that the Au 6sp intraband contributions become relatively small, whereas the contributions from the Au 5d $\rightarrow \pi^*$ (2,4-Me₂C₆H₃) and Au 6sp $\rightarrow \pi^*$ (2,4-Me₂C₆H₃) transitions are increasingly coupled with the Au 5d \rightarrow Au 6sp transitions, with an increase in excitation energies. These findings suggest that the interactions between the aromatic substituents of 2,4-DMBT ligands and the metal core have a substantial impact on the electronic and optical properties of the thiolate-protected gold cluster^{47,48}. For low-energy peak 3, however, there are more considerable contributions from Au 6sp intraband transitions in the frontier MOs, which are likely responsible for the metallic properties of **Au₁₃₈**.”

Fig. 6 | Investigation of UV-vis absorption. a UV-vis absorption spectrum (purple line) of **Au₁₃₈** in dichloromethane at room temperature⁴⁷ and the calculated spectra (red and blue lines) of the

two enantiomers with STDDFT formalism and PBE/SVP level of theory. The stick spectra are illustrated for the selected 100 transitions with highest oscillator strength. **b** Excited-state analysis for the origins of peak 1 (~360 nm), peak 2 (~430 nm), and peak 3 (~540 nm), respectively. Arrows represent the major contributions from occupied \rightarrow virtual transitions for each peak. Table S4 shows a more detailed orbital composition analysis along with transition information for the selected excited states.

Reviewer #3

(1) "... the first gold nanocluster whose inner structures are not asymmetrically induced by chiral patterns of the outermost aromatic substituents."

Concerning point 1): The surface structure of the cluster is similar to the one of $\text{Ag}_{25}(\text{DMBT})_{18}$ (the ligand is the same). Also, this cluster has a symmetric core and achiral arrangement of ligands. The NMR presented in the current paper resembles strongly the one in JPCC 2021, 125, 2524-2530.

Response: We appreciate the reviewer's comment and thank him/her for providing this Ag_{25} cluster for comparison.

1) "The surface structure of the cluster is similar to the one of $\text{Ag}_{25}(\text{DMBT})_{18}$ "

The protective shell of Au_{138} contains 24 monomeric $[-\text{SR}-\text{Au}-\text{SR}-]$ staples with four sets of two distinct C_3 -symmetric units **A** and **B** (see Fig. 3c), whereas the protective shell of $\text{Ag}_{25}(\text{DMBT})_{18}$ consists of six dimeric $[-\text{SR}-\text{Ag}-\text{SR}-\text{Ag}-\text{SR}-\text{Ag}-\text{SR}-]$ staples with a different set of assemblies (see Fig. R1). Collectively, the surface structure of Au_{138} is absolutely different from that of $\text{Ag}_{25}(\text{DMBT})_{18}$.

Figure R1. The arrangements of aryl rings in $\text{Ag}_{25}(\text{DMBT})_{18}$. **a** Three aryl rings with $\text{C}-\text{H}\cdots\pi$ interactions, two of which have $\pi-\pi$ stacking with two rings in red. **b** Three aryl rings with $\text{C}-\text{H}\cdots\pi$ interactions, only one of which has $\pi-\pi$ stacking. (Please compare to the units depicted in Fig. 3c).

2) “Also, this cluster has a symmetric core and achiral arrangement of ligands.”

Based on the detailed comparison in Figs. 2d, R1, and R2, it is clear that $\text{Ag}_{25}(\text{DMBT})_{18}$ does not possess the same degree of core symmetry as Au_{138} , due to the loss of C_3 axes. Its ligand arrangement is not achiral; the interfacial layer of $\text{Ag}_{25}(\text{DMBT})_{18}$ lacks a mirror plane.

Figure R2. Kernel structure of $\text{Ag}_{25}(\text{DMBT})_{18}$ from different views (a,b) (Please compare to the structure depicted in Fig. 2d).

Figure R3. Molecular structure and staple arrangements of $\text{Ag}_{25}(\text{DMBT})_{18}$ (a) and Au_{138} (b).

3) “The NMR presented in the current paper resembles strongly the one in JPCCC 2021, 125, 2524-2530.”

According to ^1H NMR, **Au**₁₃₈ has four sets of distinct signals from DMBT ligands with an integration ratio of 1:1:1:1, whereas $\text{Ag}_{25}(\text{DMBT})_{18}$ has only two sets of aromatic signals with a ratio of 2:1. This means that $\text{Ag}_{25}(\text{DMBT})_{18}$'s aryl rings (marked in blue in Fig. R1) with $\text{C-H}\cdots\pi$ interactions cannot be distinguished by NMR, in sharp contrast to the case of **Au**₁₃₈. The variable-temperature NMR studies of $\text{Ag}_{25}(\text{DMBT})_{18}$ do not support an interconversion process of the aryl rings in the protective shell, especially since the H_b signals do not merge as the temperature rises (see Fig. R4). Thus, the authors claimed that “The NMR investigation of $[\text{Ag}_{25}(\text{DMBT})_{18}]$ – also allowed observation of the thiolate exchange between identical nanoclusters.” In a word, their NMR spectra might be used to interpret an intermolecular transformation/decomposition but not the dynamics within the same nanocluster. The NMR presented in our work is absolutely different from the previous cases.

Figure R4. NMR studies of $\text{Ag}_{25}(\text{DMBT})_{18}$. Data from Figure 5 in *J. Phys. Chem. C* **2021**, *125*, 2524–2530 demonstrated that the proton signals at the 6-positions of 2,4-DMBT shifted in opposite directions as testing temperatures increased.

(2) First nanocluster with uncoordinated surface gold atoms.

Concerning point 2): This may be but how can the authors be sure...did they carefully check all the available structures?

Response: We thank the reviewer for his/her comment. We have carefully checked the literature and believe that **Au**₁₃₈ is the first thiolate-protected gold nanocluster with uncoordinated surface gold atoms. The first phosphine-protected gold nanocluster (**Au**₂₂) with uncoordinated surface gold atoms was reported by Lai-Sheng Wang and his co-workers (see *J. Am. Chem. Soc.* **2014**, *136*, 92–95). The binding ability of thiolate ligands to low-valent gold centers is much stronger than that of phosphine ligands; it is considerably more difficult to obtain a thiolate-protected gold nanocluster with such a coordination environment.

(3) the smallest-sized gold nanocluster having both molecular and metallic properties.

Concerning point 3): This transition is not only a function of size but also of structure. If the transition has been assigned before to about 144 gold atoms, the re-assignment to 138 atoms seems not too exciting to this reviewer.

Response: We thank the reviewer for his/her comment and agree that the transition from a molecular to a metallic state is influenced by both size and structure. However, given that all gold nanoclusters smaller than Au₁₃₃ exhibit no metal-like features, it is evident that the size effect plays a much larger role in this transition. As stated in the manuscript, there has been a long debate over whether Au₁₄₄ nanoclusters are in a metallic state (see *ACS Nano* **2021**, *15*, 13980–13992). Quan-Ming Wang and his co-workers reported in a recent publication (see *J. Am. Chem. Soc.* **2021**, *143*, 17059–17067) that Au₁₅₆ was the smallest-sized gold nanocluster with electron metallic dynamics (rather than Au₁₄₄). Based on the results of UV-vis absorption and ultrafast transient absorption, we have clearly demonstrated that **Au₁₃₈** possesses both molecular and metallic properties, which is crucial for understanding the origin of surface plasmon resonance and the nature of metallic bonds in atomically precise gold nanoclusters.

(4) The authors write “...it was realized that the NCs’ chirality might also be associated with the staple distributions or kernel structures, rather than solely being a result of chirality transfer from the external chiral organic components.”

In my opinion it is not clear if chirality is rather transferred from outside inwards or from inside outwards as a general rule. (From inside outwards sees also possible, see Dolamic, *Nature Commun.* 7117 (2015))

Response: We thank the reviewer for his/her comment. We agree that the intrinsic chirality of nanoclusters could be transferred from the inside out, and we did not rule out this possibility in the manuscript. Considering that the shell-by-shell Au₁₁₄ kernels of **Au₁₃₈** and **Au₁₄₄** are almost identical, the origin of chirality in **Au₁₄₄**’s *I*-symmetry kernel can only be explained by the “outside-in” influence from the surface and interfacial layers. If there is no such influence, the kernels would be achiral, as in **Au₁₃₈**. Since the presence of six dimeric [–SR–Au–SR–Au–SR–] staples in Au₃₈(SR)₂₄ (mentioned in *Nature Commun.* **2015**, *6*, 7117) causes the loss of mirror planes in its interfacial layer, the phenomenon of “outside-in” influence from its chiral interfacial layer providing the chirality of Au₃₈(SR)₂₄’s kernel cannot be ruled out. This argument is also supported by Professor Rongchao Jin, the leading expert in the field, who also first reported this Au₃₈(SR)₂₄ cluster (see *J. Am. Chem. Soc.* **2010**, *132*, 8280–8281). In his recent review article on Page 3 (see *Adv. Mater.* **2020**, *32*, 1905488):

“When achiral ligands are used, the origins of intrinsic chirality may come from three sources: 1) chiral kernels, e.g., Au₂₀(SR)₁₆, Au₁₃₃(SR)₅₂, etc.; 2) chiral arrangement of the staple motifs on the surface of the kernel, e.g., Au₃₈(SR)₂₄, Au₁₀₂(SR)₄₄, Au₁₄₄(SR)₆₀, etc.; 3) and the chiral whirls of carbon tails in the ligand assembly, e.g., Au₁₃₃(SR)₅₂ and Au₂₄₆(SR)₈₀. Among the three factors, the metal–ligand interface is most decisive to the intrinsic chirality of Au–thiolate NCs.”

In all previous cases, when the whirls of carbon tails are chiral, the corresponding metal–ligand interface is chiral. However, the interface is achiral in our present system, most likely due to the rapid interconversion within the dynamic chiral surface. Therefore, **Au₁₃₈** is a new type of chiral gold nanoclusters.

Reviewers' Comments:

Reviewer #1:

Remarks to the Author:

After careful review of the answer letter and the revised manuscript, I conclude that the authors have addressed most of the points raised by the referees satisfactorily. The paper is now ready for publication.

Reviewer #3:

Remarks to the Author:

The authors responded to the concerns of this reviewer, and I think that the manuscript can be published, as it is a nice contribution to the field.

The authors misunderstood the comment on the similarity with Ag₂₅. The DMBT ligand seems to favour a structure where three of these ligands interact via C-H...Pi interaction, and this interaction is observed in both clusters, independent of the different cluster underneath. This interaction is also characterized by the strongly shifted proton in the NMR (around 4.7 ppm in both cases). Pointing out this similarity in the structure (and NMR) of the ligand shell is certainly beneficial to the manuscript.

We sincerely appreciate the reviewers' time and efforts in helping us improve the quality of our work. We revised the manuscript in response to their constructive feedback and suggestions.

Reviewer #1

After careful review of the answer letter and the revised manuscript, I conclude that the authors have addressed most of the points raised by the referees satisfactorily. The paper is now ready for publication.

Response: We thank the reviewer for his/her previous insightful comments on our DFT calculations.

Reviewer #3

The authors responded to the concerns of this reviewer, and I think that the manuscript can be published, as it is a nice contribution to the field.

The authors misunderstood the comment on the similarity with Ag₂₅. The DMBT ligand seems to favour a structure where three of these ligands interact via C-H...Pi interaction, and this interaction is observed in both clusters, independent of the different cluster underneath. This interaction is also characterized by the strongly shifted proton in the NMR (around 4.7 ppm in both cases). Pointing out this similarity in the structure (and NMR) of the ligand shell is certainly beneficial to the manuscript.

Response: We value the reviewer's feedback and have revised the manuscript accordingly:

“Since there is no π – π stacking within the *trans* [–SR–Au–SR–] staples of protective unit **B**, for the thiolate primarily with C–H $\cdots\pi$ interactions, the proton at the 5-position appears at 4.73 ppm, the second most upshifted signal in **Au₁₃₈**. This shielding effect is also observable in other nanocluster systems containing 2,4-DMBT ligand units assembled in a comparable manner⁴⁰.”

We cited the reference mentioned by this reviewer (see Ref 40: *J. Phys. Chem. C* **2021**, *125*, 2524–2530).